# Junior to senior transition of male elite junior tennis players: A retrospective study

**Michal Bozděch** [ID]**[1]\*, Jiří Zháněl** [ID]**[2]**

**1** Department of Physical Education and Social Sciences, Faculty of Sports Studies, Masaryk University, Brno, Czech Republic, **2** Department of Sport Performance and Exercise Testing, Faculty of Sports Studies, Masaryk University, Brno, Czech Republic

\* michal.bozdech@fsps.muni.cz

**Data Availability Statement:** All relevant data are within the paper and its Supporting Information files (S1).

**Funding:** JZ was funded by The Masaryk University (Specific research) for the project entitled Laterality

## Abstract

This study explores the intricate dynamics of the Junior-to-Senior (JTS) transition phase in elite tennis. Focusing on challenges faced by young talents, the research aims to unveil factors influencing successful transitions and the role of elite junior tournaments. In a retrospective analysis, male tennis players (n = 240) from national teams in the ITF World Junior Tennis Finals tournament (2012–2016) were analyzed using Chi-square tests, Cramer's V, Bayesian statistics, and Multinomial Logistic Regression (MLR). Results revealed that 62.1% of elite junior participants were found in the Association of Tennis Professionals database, emphasizing the significance of team nominations and tournament results as important variables to monitor. Inferential and Bayesian statistics confirmed robustness, with MLR highlighting tournament results' importance. The findings highlight the complexities of the JTS transition and underscore the pivotal roles of participation, national team nominations, and tournament results. The study recommends the implementation of comprehensive player development programs, urging strategic team selections by national federations and academies. Coaches, stakeholders, and organizations should prioritize monitoring these variables for early talent identification and support. These measures collectively aim to optimize success trajectories, navigating the critical JTS phase in junior tennis players' sporting careers.

## Introduction

Tennis is one of the most popular sports globally, known for its rigorous demands on physical fitness, technical prowess, and psychological resilience [1, 2]. The preparation for a professional tennis career typically begins at a young age due to the extended time required to develop essential skills and capabilities.

Not only in tennis but also across all sports, there is an urgent need to scrutinize the transition from junior to professional categories for both genders and address issues such as burnout, transitions, or dual careers [2–5]. The decline in active physical activity among adolescents [8, 9] has led to an increased emphasis on active and high-quality monitoring for those aspiring to maintain an active lifestyle. Consequently, alongside monitoring

in the context of diagnosis of selected factors of sports performance in tennis and injury prevention (Grant No. MUNI/A/1637/2020). The full name of the funder is Masaryk University, and more information can be found on their website: https://www.muni.cz/en/research/projects/59488. The funders had no role in the study design, data collection and analysis, decision to publish, or preparation of the manuscript.

**Competing interests:** The authors have declared that no competing interests exist.

performance characteristics, there is a growing focus on studies that delve into the junior-to-senior transition (JST), recognized as a pivotal phase in the development of young, promising, or elite athletes [6]. This scrutiny allows for a better understanding, prediction, or reduction of burnout or premature termination of a sports career before reaching peak performance, often associated with a disdain for physical activity [7, 8].

The JST phase is crucial in the development of young, promising, or elite athletes, signifying the shift from junior/youth age categories to the professional level [9]. Typically lasting between one and four years, this phase is characterized by non-linear processes, presenting a dynamic and complex developmental stage accompanied by sociocultural barriers and heightened expectations [10–12]. One of the primary factors in this transition is the need for a structured training program that emphasizes both physical preparation and sport-specific skills [13], which must prioritize health first [14]. Effective periodization of training is essential to optimize performance while minimizing the risk of injury and burnout. This involves breaking down training into distinct phases, including general preparation, specific preparation, and competition phases, with active recovery periods to alleviate fatigue and maintain fitness levels [15]. Training preparation and participation in prestigious international tournaments can be considered an adequate opportunity for the development of the experiences and skills that tennis players will regularly utilize throughout their careers.

Athletes navigate this transitional phase amidst elevated psychological stress, emotional, moral, and performance-related standards, making it the most challenging and critical period in their sporting careers [9, 16, 17]. As young tennis players move from youth competitions to the more demanding professional circuit, key aspects influencing the JTS transition include physical development, mental resilience, skill refinement, competitive experience, and support systems from stakeholders such as coaches, family, scouts, and organizations [14, 18]. The professional tennis circuit, lasting 11 months of the year, takes elite junior players from the relative comfort of top-level junior tournaments to low-level senior tournaments with limited prize money, difficult playing conditions, and often reduced support due to a lack of funding, availability of appropriate role models, and extensive travel. This background creates an appropriate context for intervention, offering coaches, sports scientists, and organizations the opportunity to support junior athletes in their preparation for and navigation of the transition [15].

Elite players strive to overcome this phase with the least risk of injury and negative consequences [9, 16, 17, 19]. Consequently, the dropout rate among junior athletes during this phase can be as high as two-thirds [18]. While temporary biological advantages related to birth dates may aid relatively older peers, evidence indicates a higher dropout rate among those who unknowingly benefited from this advantage, particularly after adolescence and within two years at the professional level; often before reaching their peak performance [8, 20–23].

The complexity of tennis, akin to most high-performance sports, involves multifaceted associations and covariations impacting the course and outcome of the game, tournament, season, health, JTS transition, and the entire sports career. These associations encompass performance-based as well as socio-cultural, financial, technological, and stakeholder support elements [24, 25]. However, significant research gaps remain in understanding the specific strategies that can facilitate this transition effectively. Further studies are needed to explore best practices for current athletes and to develop evidence-based recommendations for coaches and sports organizations to support young athletes as they progress to senior levels of competition.

Therefore, this study aims to analyze the probability of elite junior tennis players entering the professional tennis association. This retrospective-predictive study aims to develop a valid model for predicting whether participants in elite junior tournaments will become professional tennis players.

## Methods

### Participants

The World Junior Tennis Finals (WJTF) is a prestigious team tournament for players aged 14 and under. The tournament has been organized by the International Tennis Federation (ITF) since 1991 and, since 1999, in collaboration with the Czech Tennis Association and the tennis club TK PLUS in Prostějov. This competition gathers the top junior boys and girls from around the world, whose teams qualify based on results from regional tournaments, offering them a competitive platform that prepares them for a professional career. Many participants in this tournament have historically transitioned to successful professional careers, including notable players such as Rafael Nadal, Novak Djokovic, Carlos Alcaraz, and Tomáš Berdych. The tournament provides young athletes with invaluable experience and opportunities to showcase their skills on a global stage. This underscores the significance of the WJTF in our study, as it serves as an ideal setting to analyze the potential of elite junior players advancing to professional levels. Tennis players from national representative teams who advanced to The WJTF tournament between 2012 and 2016 were participants in this retrospective-predictive study. This time frame was intentionally selected to allow participants the opportunity to reach their peak performance age (around 24 years) during the data collection year [25, 26]. The research exclusively focused on male tennis players to ensure study accuracy and minimize potential gender-related influences. Previous studies have identified significant gender-related differences in performance-related variables [19]. Including both male and female players could introduce heterogeneity in the data, potentially confounding the results and leading to less reliable conclusions. By focusing solely on male players, the study aims to provide a clearer understanding of the factors influencing performance within a more homogeneous group [28]. Out of the total 240 tracked players, eight participated in the elite tournament more than once. It is important to note that the results of the pilot study were presented at a conference. However, different statistical methods were applied for this research to confirm a different research intent and objectives. The research adhered to the principles outlined in the Declaration of Helsinki and received approval from the Masaryk University Research Ethics Committee (EVK-2021-006). Even if the data was anonymized after completion. Due to the design and objectives of this study, the authors were able to identify individual participants. However, due to the retrospective nature of this research, this did not affect the results but served to provide a more detailed understanding of the investigated effects.

### Data collection

Researchers obtained foundational data and consent to use it from tournament organizers, consisting of official materials, for each year of the elite tournament. Subsequently, these materials were supplemented with information related to the sports career, acquired from the official publicly accessible Association of Tennis Professionals (ATP) website (https://www.atptour.com/en/). Specifically, players were individually searched and classified by ATP status in single; Not Found (NF), Registered (R) without points, and Found (F). For Found tennis players who accumulated sufficient points from ATP tournaments, the value of their best career ranking was also obtained (updated as of the last week of the 2022 season; data collection from the professional association took place retrospectively in the first quarter of 2023). Data were validated and cleaned of outliers.

### Research criteria

To address the research objectives, the data was categorized according to the following criteria: *ATP Status (F/NF)* [Found or Not Found in ATP database; dichotomous data], *ATP Status (F/*

*R/NF)* [Registered without points, Found and Not Found in ATP database; nominal data), *Nomination* [1–3 rank, a team in an elite junior tournament consists of 3 tennis players with 3 predefined levels, often determined according to the current national ranking; ordinal data], *Year of the tournament* [2012–2016 year, five consecutive tournaments from 2012 through 2016; nominal data], *Continent* [countries represented by individual tennis teams were converted into continents based on their predominant geographical location; nominal data]; *Birth quarter* [$Q_1$-$Q_4$, obtained from the birth date precisely, the month which was then converted into quarters, January-March, April-June, etc.; nominal data]; *WJTF results* [1–4 category, each year 16 teams participated in the tournament, data was converted into quarters, $1^{st}$-$4^{th}$ place, $5^{th}$-$8^{th}$ place, etc.; ordinal data].

## Statistical analysis

The statistical analysis chosen to address the research objectives initially included methods and procedures from inferential statistics (Chi-square; $\chi^2$), Effect Size (Cramer's V), Bayesian approach, Regression analysis (Multinomial logistic regression, MLR).

This approach was chosen to support complementarity between Inferential statistics, Effect Size, and the Bayesian approach [27]. Chi-square ($\chi^2$) test of independence, Cramer's V test with Cohen [28] interpretation, and Bayesian statistics ($BF_{10}$ and $BF_{01}$) were utilized to evaluate the different frequencies among participants in the elite tournament who were Found or Not Found in the ATP database depending on various research variables (Nomination, Tournament year, Continent, Birth quartile, WJTF results). A priori Power analysis for the Chi-square test was conducted considering the expected effect size, level of significance, and test power, utilizing G*Power 3.1.9.6 software. For a medium effect with $\alpha$ = 0.05 and 1-$\beta$ = 0.95, a total sample of 172 participants was required, based on the values derived from the WJTF results variable. With the current participant count in the study (*n* = 240) at $\alpha$ = 0.05, the Power (1-$\beta$) exceeds 0.99. The most prominent variables from this phase (Nomination and WJTF results) were further analyzed using MLR where the dependent variable was ATP status (the reference category being Not Found in the ATP database) and the factors were Nomination and WJTF results. Out of 240 observations, no data were missing. The Likelihood Ratio chi-square test was used to compare the full model against a null or no predictors model. The final model showed a significant improvement in fit over a null model; $\chi^2(5)$ = 11.68, *p* = .039, indicating that the full model represents a significant improvement in fit over the null model. The Goodness of Fit test indicated that both Pearson's [$\chi^2(6)$ = 4.79, *p* = .571] and Deviance [$\chi^2(6)$ = 5.01, *p* = .543] Chi-square tests indicated a good fit. The statistical analysis was conducted using IBM SPSS Statistics (Version 29.0.0, IBM Corp, Armonk, NY) and JASP (Version 0.18.3, University of Amsterdam) software packages for Windows. Statistical significance was assessed using a significance level (α) of 0.05. Therefore, results with a p-value less than 0.05 were considered statistically significant.

## Results

Among the 240 participants in the elite junior tournament, 195 were identified in the official ATP database. Of these, 46 were merely registered without acquiring any points, while 149 (62.08%) earned official ranking points. Among these 149 players, 64 (42.95%) achieved a position within the top 500 in their tennis careers, 40 (26.85%) were ranked between 501–1000, 29 (19.46%) between 1001–1500, and 16 (10.74%) attained rankings of 1501 and above. This data indicates a trend where more frequent participation in elite tournaments correlates with achieving higher career rankings (Table 1).

**Table 1. An overview of ATP status distribution according to research variables.**

| Variable | | Players ATP status, $n$ (%) | | $n$ | $p$ | V | BF$_{10}$ | BF$_{01}$ |
|---|---|---|---|---|---|---|---|---|
| | | Not found | Found | | | | | |
| Nominations in team | 1st | 25 (10.42 %) | 55 (22.92 %) | 80 | .024 | .176* | 1.58 | 0.63 |
| | 2nd | 26 (10.83 %) | 54 (22.50 %) | 80 | | | | |
| | 3rd | 40 (16.67 %) | 40 (16.67 %) | 80 | | | | |
| Year of the tournament | 2012 | 19 (15.83 %) | 29 (24.17 %) | 48 | .435 | .126* | 0.03 | 32.83 |
| | 2013 | 14 (11.67 %) | 34 (28.33 %) | 48 | | | | |
| | 2014 | 17 (14.17 %) | 31 (25.83 %) | 48 | | | | |
| | 2015 | 18 (15.00 %) | 30 (25.00 %) | 48 | | | | |
| | 2016 | 23 (19.17 %) | 25 (20.83 %) | 48 | | | | |
| Continent | Africa | 12 (13.19 %) | 6 (6.59 %) | 18 | .070 | .206** | 0.15 | 6.51 |
| | Asia | 22 (24.18 %) | 29 (31.87 %) | 51 | | | | |
| | Australia | 4 (4.40 %) | 8 (8.79 %) | 12 | | | | |
| | Europe | 24 (26.37 %) | 60 (65.93 %) | 84 | | | | |
| | N. America | 12 (13.19 %) | 18 (19.78 %) | 30 | | | | |
| | S. America | 17 (18.68 %) | 28 (30.77 %) | 45 | | | | |
| Birth quarter | $Q_1$ | 52 (21.67 %) | 72 (30.00 %) | 124 | .312 | .122* | 0.04 | 22.66 |
| | $Q_2$ | 25 (10.42 %) | 39 (16.25 %) | 64 | | | | |
| | $Q_3$ | 9 (3.75 %) | 25 (10.42 %) | 34 | | | | |
| | $Q_4$ | 5 (2.08 %) | 13 (5.42 %) | 18 | | | | |
| WJTF results | 1.-4. place | 13 (5.42 %) | 47 (19.58 %) | 60 | .002 | .251** | 24.5 | 0.04 |
| | 5.-8. place | 20 (8.33 %) | 40 (16.67 %) | 60 | | | | |
| | 9.-12. place | 25 (10.42 %) | 35 (14.58 %) | 60 | | | | |
| | 13.-16. place | 33 (13.75 %) | 27 (11.25 %) | 60 | | | | |
| Total | | 91 (37.92 %) | 149 (62.08 %) | 240 | | | | |

$Q_i$ birth quarter.

*small ES.

**Medium ES.

***Large ES. BF$_{10}$ Bayes factor in favor alternative hypothesis over null hypothesis. BF$_{01}$ Bayes factor in favor null hypothesis over alternative hypothesis.

The Chi-square test revealed significant differences in ATP status proportions across different levels of team nominations ($\chi^2(2) = 7.47$, $p = .024$) and WJTF results (place) ($\chi^2(3) = 15.06$, $p = .002$). However, after Bonferroni correction ($p_{adj} = .001$), only WJTF results remained statistically significant. These results suggest that better team performance in elite tournaments is associated with a higher likelihood of players entering the professional senior category. Specifically, players from better-performing teams had a higher probability of reaching the ATP rankings. To further clarify the findings, the study used verbal classification of effect sizes: trivial, small, medium, and large. A small to medium effect size was identified, indicating a moderate association between ATP status and both team nomination and WJTF results. The medium effect is noticeable for variables that were statistically significant (based on the $\chi^2$ test), indicating a moderate association between Players ATP status and Nomination in the team or results at the elite junior tennis tournament.

Bayesian statistics provided additional insight, showing that the likelihood of differences between variables was significantly higher, particularly for WJTF results, which were 1.6 to 24.5 times more likely to be associated with ATP status differences than not. This finding emphasizes the substantial influence of WJTF results on a player's future career trajectory.

In Fig 1 illustrates the association between team nomination and tournament results, with a regression line and a 95% confidence interval. The data indicates that lower team nominations

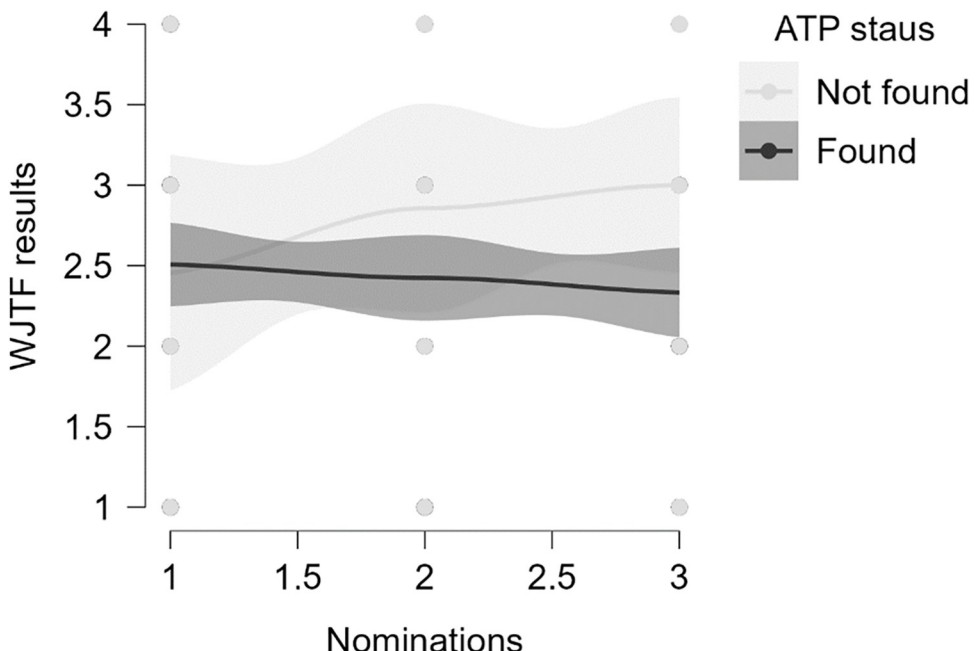

**Fig 1. An overview of association between nomination in team and tournament results.**

correlate with poorer performance in junior tournaments, which increases the likelihood of early career termination.

The MLR analysis further supported these findings. It showed that WJTF results were a significant predictor of ATP status (p = .041), whereas team nomination was not statistically significant (p = .17). The analysis revealed that players with worse WJTF results had a higher likelihood of being found in the ATP database, with an Odds Ratio ranging from 2.67 to 3.07, indicating a strong association.

These results underline that WJTF performance is a crucial factor in predicting a young tennis player's future professional trajectory, with players from lower-ranking teams surprisingly showing a higher likelihood of ATP registration. This counterintuitive finding suggests that factors other than immediate tournament success, such as long-term development and support, may play a critical role in a player's progression to professional levels. However, it is important to note that the conclusion suggesting poorer WJTF tournament placement associated with better ATP ranking was evident only from the MLR analysis, not from the other analyses. Further research is needed to validate these conflicting conclusions.

## Discussion

The present study underscores the formidable challenges inherent in the transition from elite junior tennis to the ATP, emphasizing the pivotal role of the JST phase. To enhance the prospects of aspiring players, national tennis federations and academies should implement comprehensive player development programs that prioritize both individual performance and strategic team dynamics. The findings highlight the importance of participating in an elite junior tournament, as evidenced by the unexpectedly high percentage of junior elite tennis players achieving professional ranking status. Additionally, the significance of strategic team nominations was demonstrated, showing that teams with higher nominations, even without securing victory, exhibited a heightened likelihood of successful transitions to the ATP. The

relevance of tournament results was also evident; descriptive statistics indicated that the relative number of tennis players who made it to the professional level increased with better team positions. However, for those who did not make it to the ATP, this number decreased with worse positions. Inferential statistics ($\chi^2$, ES, BF) confirmed these trends, although MLR analysis suggested a slight, albeit negligible, trend in which players with poorer team positions had a greater chance of entering the ATP. These discrepancies in conclusions likely stem from the application of different statistical methods, which address distinct questions in varying ways.

Moreover, a similar study focusing on elite junior female tennis players transitioning to the Women's Tennis Association (WTA) found that out of 240 participants in an elite junior tournament, 141 (58.8%) gained points in the WTA rankings [29]. This is eight participants fewer (3.3%) than reported in the current study ($n$ = 149, 62.1%). Both studies revealed a lower dropout rate (girls = 41.3%, boys = 37.9%) than reported by Franck et al. [18], who noted up to a 75.0% dropout rate among junior athletes. This suggests that participation in elite junior tournaments plays a crucial role in the JTS transition. However, it is essential to note that other important factors were not examined within the scope of this study, such as skills, psychological factors, health status, and socio-cultural and economic factors, among others.

The study indicates that participants in elite junior tournaments commonly achieve rankings in the 201–300 range in their best career ATP ranking. This finding aligns with Balliauw et al. [33], who asserted that professional tennis players ranked below the top 250 face financial challenges, potentially influencing the continuation of their sports careers. Therefore, it can be stated that this ranking range represents an important professional threshold, and it is crucial for coaches, players, associations, and stakeholders to be aware of this.

### Limitations and directions for future research

Despite the diligent efforts of the researchers, this study has its limitations. The focus on participants in elite competitions may overlook valuable information about those who did not reach the finals but performed well in semi-finals or at the state level, just below the team selection threshold. Expanding the dataset could enhance the understanding of the JTS process, but it would not provide new data on tournament nominations and results from the elite junior tournament, which were identified as the most important variables.

One potential limitation is that our primary data, obtained from an elite junior tennis tournament, were combined with information from a secondary source (the official ATP website). However, this is a common approach to data acquisition and analysis, as demonstrated by an extensive study by Takahashi et al. [29], who identified six common methods for obtaining tennis data (data mining, tracking, video recording, coaches, broadcasting, and the Internet). Of the 90 different studies examined, 38 utilized a similar approach, using secondary data from publicly available web sources. This practice is standard in the sports research environment. Nevertheless, to determine causal relationships, it would be advisable to conduct a longitudinal study, ideally in collaboration with sports associations, coaches, stakeholders, and other researchers.

The current study exclusively focused on male tennis players, excluding their female counterparts due to substantial disparities observed in performance and strategy. These distinctions are not only evident across the spectrum of professional male and female players but also between junior and professional male players [30–33]. This highlights the need for gender-specific as well as junior and senior-specific training, emphasizing the importance of adopting distinct training methods and customizing approaches based on performance levels. It also underscores the essential development of suitable methods tailored for monitoring each respective sex and phase.

## Conclusion

This study explores the challenges associated with transitioning from elite junior to professional tennis, known as the JST transition. Key findings emphasize the predictive power of participation in elite junior tennis tournaments, with 62.1% of elite junior participants advancing to actively participate in the ATP.

Better rankings at the elite junior tournament are linked to a higher likelihood of being listed in the ATP ranking database and achieving successful professional careers. The study identifies team nominations and, more significantly, tournament results as crucial predictors of future professional status.

To enhance support for athletes during the challenging JST phase, the establishment of a robust longitudinal monitoring system is recommended. Such a system, facilitated by collaborative efforts among tennis organizations, coaches, stakeholders, and sports scientists, would enable comprehensive tracking of players' progress and provide timely interventions and support. Additionally, educational programs addressing the unique stressors and challenges associated with the JST phase are crucial for preparing athletes for the demands of professional tennis. Coaches, scouts, and tennis organizations should prioritize monitoring elite junior tournament results to identify emerging talent early. Overall, a holistic approach to player development, strategic team selections, and the judicious use of predictive analytics are essential for optimizing the career trajectories of junior tennis players as they transition to the professional circuit.

## Supporting information

**S1 File.**
(PDF)

## Author Contributions

**Conceptualization:** Michal Bozděch.

**Data curation:** Jiří Zháněl.

**Formal analysis:** Michal Bozděch.

**Funding acquisition:** Jiří Zháněl.

**Investigation:** Michal Bozděch.

**Methodology:** Michal Bozděch.

**Project administration:** Jiří Zháněl.

**Resources:** Jiří Zháněl.

**Software:** Michal Bozděch.

**Supervision:** Michal Bozděch.

**Visualization:** Michal Bozděch.

**Writing – original draft:** Michal Bozděch.

**Writing – review & editing:** Michal Bozděch, Jiří Zháněl.

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
