## [Decision Letter · Decision Letter 0]

9 Jul 2024

PONE-D-24-14066Junior to senior transition of male elite junior tennis players: a retrospective studyPLOS ONE

Dear Dr. Bozděch,

Thank you for submitting your manuscript to PLOS ONE. After careful consideration, we feel that it has merit but does not fully meet PLOS ONE’s publication criteria as it currently stands. Therefore, we invite you to submit a revised version of the manuscript that addresses the points raised during the review process.

The reviewers have raised several critical points that need to be addressed to improve the clarity, coherence, and overall quality of your paper. The main issues highlighted include the clarity and relevance of the conclusions drawn from the results, the complexity and variety of statistical analyses used, and several language-related problems, including unclear wording and unnecessary capitalization. Additionally, there are concerns regarding the methodological approach and the exclusion of female players from the study, which need to be thoroughly justified.

To address these concerns, please focus on the following key areas in your revision. Firstly, ensure that the results and conclusions are directly aligned, and clarify how your findings relate to psychosocial stressors and pressures during the junior to senior transition in tennis. Simplify the statistical analyses to enhance readability and focus on the most critical results that support your conclusions. Additionally, review the manuscript for language clarity, correct any unclear wording, and remove any unnecessary capitalization. Justify the rationale for excluding female players and consider discussing the implications of this choice. Lastly, please incorporate the specific comments and questions highlighted in the attached PDF from the reviewer and seek the assistance of a native English speaker to polish the language. 

We look forward to receiving your revised manuscript.

Kind regards,

Gábor Vattay, PhD, DSc

Academic Editor

PLOS ONE

Reviewers' comments:

Reviewer's Responses to Questions

**Comments to the Author**

1. Is the manuscript technically sound, and do the data support the conclusions?

Reviewer #1: No

Reviewer #2: Yes

2. Has the statistical analysis been performed appropriately and rigorously? 

Reviewer #1: No

Reviewer #2: Yes

3. Have the authors made all data underlying the findings in their manuscript fully available?

Reviewer #1: Yes

Reviewer #2: Yes

4. Is the manuscript presented in an intelligible fashion and written in standard English?

Reviewer #1: No

Reviewer #2: No

5. Review Comments to the Author

Reviewer #1: Thank you for the opportunity to review your manuscript. At present, I do not feel the results support some of the conclusions in the discussion of the paper. In particular, I struggled to understand how the reported findings related to psychosocial stressors and pressures during the junior to senior transition. I also felt the varied statistics used detracted from the results, as this created some confusion when reading the manuscript and made it difficult to know which results were most important for my interpretation as a reader. I also wish to note a few issues with the language throughout, including unclear language and some words that were capitalised and did not need to be, which require revisions.

Reviewer #2: The reviewer would like to thank the authors for doing research in such an interesting field for the sport of tennis.

The following comments and questions are made with the intention of assisting the authors in improving the quality of the manuscript which is already quite high. Therefore, the reviewer would like the authors to understand these comments as part of the scientific debate that is needed for the advancement of science.

To facilitate this process, the reviewer has added the pdf file with highlighted sections, comments, and questions for the authors to address.

The reviewer believes that if the authors address these comments and make minor changes to the manuscript, it should be accepted for publication.

Finally, the reviewer suggests that the authors seek for the assistance of a native English speaker person to improve some sections of the manuscript.

6. PLOS authors have the option to publish the peer review history of their article (what does this mean?). If published, this will include your full peer review and any attached files.

Reviewer #1: No

Reviewer #2: **Yes: **Miguel Crespo

---

## [Author Response · Author response to Decision Letter 0]

30 Jul 2024

Thank you very much for your time and willingness to devote to our manuscript. We believe that the article is better because of you.

---

## [Editor Report · Decision Letter 1]

31 Jul 2024

PONE-D-24-14066R1Junior to senior transition of male elite junior tennis players: a retrospective studyPLOS ONE

Dear Dr. Bozděch,

Thank you for submitting your manuscript to PLOS ONE. After careful consideration, we feel that it has merit but does not fully meet PLOS ONE’s publication criteria as it currently stands. Therefore, we invite you to submit a revised version of the manuscript that addresses the points raised during the review process.

While you sent a revised version of the manuscript, we didn't receive a detailed description of the changes made and the replies to the reviewers comments as required. Please, repeat the submission and include all the necessary parts!

We look forward to receiving your revised manuscript.

Kind regards,

Gábor Vattay, PhD, DSc

Academic Editor

PLOS ONE

---

## [Author Response · Author response to Decision Letter 1]

2 Aug 2024

We sincerely thank the reviewers and editor for their time and effort to improve this manuscript. We believe that thanks to you, our article is now significantly better.

---

## [Decision Letter · Decision Letter 2]

13 Aug 2024

Junior to senior transition of male elite junior tennis players: a retrospective study

PONE-D-24-14066R2

Dear Dr. Bozděch,

We’re pleased to inform you that your manuscript has been judged scientifically suitable for publication and will be formally accepted for publication once it meets all outstanding technical requirements.

Kind regards,

Gábor Vattay, PhD, DSc

Academic Editor

PLOS ONE

Additional Editor Comments (optional):

Reviewers' comments:

Reviewer's Responses to Questions

**Comments to the Author**

1. If the authors have adequately addressed your comments raised in a previous round of review and you feel that this manuscript is now acceptable for publication, you may indicate that here to bypass the “Comments to the Author” section, enter your conflict of interest statement in the “Confidential to Editor” section, and submit your "Accept" recommendation.

Reviewer #2: All comments have been addressed

2. Is the manuscript technically sound, and do the data support the conclusions?

Reviewer #2: Yes

3. Has the statistical analysis been performed appropriately and rigorously? 

Reviewer #2: Yes

4. Have the authors made all data underlying the findings in their manuscript fully available?

Reviewer #2: Yes

5. Is the manuscript presented in an intelligible fashion and written in standard English?

Reviewer #2: Yes

6. Review Comments to the Author

Reviewer #2: The reviewer would like to thank the authors for their understanding and for accepting the suggestions made in the first review.

As the authors indicate, he considers that the manuscript has improved considerably and therefore suggests that it be accepted for publication.

Finally, the reviewer would like to thank the authors for their interest in publishing their research on tennis and on such an important topic as the one addressed in this research.

7. PLOS authors have the option to publish the peer review history of their article (what does this mean?). If published, this will include your full peer review and any attached files.

Reviewer #2: **Yes: **Miguel Crespo

---

## [Editor Report · Acceptance letter]

22 Aug 2024

PONE-D-24-14066R2 

PLOS ONE

Dear Dr. Bozděch, 

I'm pleased to inform you that your manuscript has been deemed suitable for publication in PLOS ONE. Congratulations! Your manuscript is now being handed over to our production team.

Kind regards, 

on behalf of

Dr. Gábor Vattay 

Academic Editor

PLOS ONE